# The Sequencing-Based Mapping Method for Effectively Cloning Plant Mutated Genes

**DOI:** 10.3390/ijms22126224

**Published:** 2021-06-09

**Authors:** Li Yu, Yanshen Nie, Jinxia Jiao, Liufang Jian, Jie Zhao

**Affiliations:** State Key Laboratory of Hybrid Rice, College of Life Sciences, Wuhan University, Wuhan 430072, China; 2012102040065@whu.edu.cn (L.Y.); nie_ys@whu.edu.cn (Y.N.); 2016202040087@whu.edu.cn (J.J.); jianliufang@whu.edu.cn (L.J.)

**Keywords:** map-based cloning, whole-genome sequencing, heterozygote, rice

## Abstract

A forward genetic approach is a powerful tool for identifying the genes underlying the phenotypes of interest. However, the conventional map-based cloning method is lengthy, requires a large mapping population and confirmation of many candidate genes in a broad genetic region to clone the causal variant. The whole-genome sequencing method clones the variants with a certain failure probability for multiple reasons, especially for heterozygotes, and could not be used to clone the mutation of epigenetic modifications. Here, we applied the highly complementary characteristics of these two methods and developed a sequencing-based mapping method (SBM) for identifying the location of plant variants effectively with a small population and low cost, which is very user-friendly for most popular laboratories. This method used the whole-genome sequencing data of two pooled populations to screen out enough markers. These markers were used to identify and narrow the candidate region by analyzing the marker-indexes and recombinants. Finally, the possible mutational sites were identified using the whole-genome sequencing data and verified in individual mutants. To elaborate the new method, we displayed the cloned processes in one *Arabidopsis* heterozygous mutant and two rice homozygous mutants. Thus, the sequencing-based mapping method could clone effectively different types of plant mutations and was a powerful tool for studying the functions of plant genes in the species with known genomic sequences.

## 1. Introduction

In the studies of plant genes, the mutant is a very important and effective tool for it is the direct embodiment of loss and the most convincing evidence of gene function. To clone the mutated genes, many methods have been invented and used. For example, Genome Walking and Tair-PCR (Thermal Asymmetric Interlaced PCR) were employed to identify the mutated sites induced by the insertion of T-DNA, and TILLING (Targeting Induced Local Lesions IN Genomes) was invented for the cloning of the mutational sites induced by EMS (Ethyl methyl sulfonate) [1,2,3]. The most popular methods are the conventional map-based cloning method and the whole-genome sequencing method, including SHOREmap and MutMap [4,5,6,7].

The conventional map-based cloning methods are illustrated in Appendix A and could be used for all types of mutation, including DNA deletion, DNA insertion, the mutation of a single nucleotide and the mutation of epigenetic modifications (Appendix A) [4]. For good effect, it needs a good mapping population that can be used to screen out enough markers and that is big enough to get the lines that harbor the crossover near the mutated site [8,9,10,11,12]. However, the available markers are usually different between the different hybrid combinations and difficult to identify, using the conventional method [13,14]. Many plants, especially for crops like rice and maize, need more space to plant the large mapping populations, which is very difficult for many laboratories [11,15,16]. Besides, hybrid plants, such as rice, either display serious abortion in F_1_ generation, resulting in insufficient numbers of F_2_ populations, or have fewer SSR markers, making it difficult to meet the requirements of map-based cloning [17,18,19]. Moreover, the target region is usually about 30–500 kb using the conventional map-based cloning method and there are about 5–100 candidate genes for verification and complementation which take a long time [16,17]. These restrict the application of conventional map-based cloning methods. The whole-genome sequencing method was invented to clone the mutations using next-generation sequencing (NGS) technology which provides an unprecedented wealth of high-resolution genotypic information. Thus, the whole-genome sequencing method overcomes many traditional difficulties, such as time-consuming genetic assays and a big planting space. Based on the whole-genome sequencing method, many different strategies have been invented to analyze the sequencing data and to identify causative mutations more effectively, like SHOREmap, NGM (next-generation mapping) and MutMap [5,6,7]. In spite of this, there are many unassociated polymorphisms which segregate with the causative mutation, sequences different from the ‘reference genome’, and biases of the genome sequencing for their sequencing depth is generally no more than 40×, leading to some large gaps and a low signal-to-noise ratio. To overcome these problems, either multiple backcross with parents, or bulk analysis of a very large number of mutant lines with high depth, or genome sequencing of many individual plants from the same mutant, or a de novo assembly of the genomic gap region are performed to improve cloning efficiency, however, the cost also increases which is not favorable for the ordinary small laboratories [7,20,21,22,23]. Even if these are achieved, it is still difficult to clone the target genes in the species with complex and large genomes for the existence of the repetition of large fragments. Besides, many homozygous mutants could not survive for some genes were indispensable for the survival of sporophytes or gametophytes, thus only heterozygotes could be used to clone the target genes. The heterozygotes harbor smaller marker indexes around the mutated sites compared to the homozygotes and produce lower signal-to-noise ratio, making it more difficult to clone the mutational sites using the whole-genome sequencing method. Moreover, the whole-genome sequencing method could not be used to clone the mutation of epigenetic modification for there is no difference of sequence in these mutants. Thus, there is no heterozygous or epigenetic mutant that has been cloned by the whole-genome sequencing method, and a more effective and simple method is needed to address these issues.

Here, we screened out some rice and *Arabidopsis* mutants, and planned to use the whole-genome sequencing method to clone the mutated genes. However, we failed to get the target genes of five mutants and only got the mutated gene of one *Arabidopsis* homologous mutant. To clone the target genes, we improved the conventional map-based cloning method and introduced the whole-genome sequencing technology to generate a new method—the sequencing-based mapping method (SBM). We used the whole-genome sequencing data to screen out enough markers, and then, the marker-indexes and recombinants of these phenotype-linked markers were analyzed to identify and narrow the candidate region. Finally, possible mutational sites were identified using the whole-genome sequencing data and verified in the mutant. Based on this new method, the mutated sites of three rice homologous mutants and two *Arabidopsis* heterozygous mutants were cloned using a small mapping population. The cloning procedures of one *Arabidopsis* heterozygous mutant and two rice homologous mutants are shown here to illustrate this method.

## 2. Results

### 2.1. Cloning Procedure of Mutant Genes by Using SBM

The sequencing-based mapping method (SBM) was based on the whole-genome-sequencing technique and used to clone the mutants in the species with known genomic information. This method could be used to clone all types of mutations efficiently, especially the heterozygous mutants. The procedure of the method in heterozygous and homologous mutants of the diploid species consists of four steps and is described in Figure 1 and Appendix A respectively.

Step One: Construction of the mapping population. The heterozygous mutants that could not produce homozygous mutants and their heterozygotes exhibited a certain abortion rate of seeds (about 0.25–0.5). This suggested that the heterozygotes should be crossed with WT (wildtype) of the closely related variety for the offspring obtained from the cross of distant varieties exhibited a certain abortion rate of seeds. For example, the cross between *Arabidopsis* ecotype Columbia (Col) mutant and Landsberg *erecta* (L*er*), or between rice *japonica* varieties, could produce F_1_ plants that showed almost no abortion. In the heterozygotes, half of the target genes were mutated and their marker indexes were 0.5, therefore, we chose BC_3_F_1_ population (the marker indexes of the phenotype-linked markers were about 0.5 while the marker indexes of the phenotype-unlinked markers were about 0.0625), but not the F_2_ population (the marker indexes of all the markers were about 0.5), as the mapping population (Figure 1).

Step Two: Whole-genome sequencing. About 30–200 heterozygous plants of BC_3_F_1_ population were selected to extract their DNAs, mixed with an equimolar of each, resequenced, and analyzed. The SNP (single nucleotide polymorphism), InDel (insertion and deletion), structural variations (SV, mainly insertion and deletion of large fragments) and copy number variation (CNV) were identified and analyzed to calculate their marker index, the rate of one parent-type allele. The high-quality alleles, whose supported reads were more than eight, were identified and used to calculate their Δmarker-index (mainly SNP and InDel). The Δmarker-index plots were drawn and the average distributions of the Δmarker index were produced using the MutMap method to identify the candidate regions and sites (Figure 1). If the numbers of the candidate mutated sites were very few and the mutated gene was verified among them by the DNA-sequencing technology, then the cloning experiments could be finished. However, all the heterozygous mutants failed to produce the target genes by only using the whole-genome sequencing method. Thus, more work should be done to clone the target genes.

Step Three: Identification and analysis of phenotype-linked markers. More than 30 heterozygous plants of the BC_3_F_1_ population were used for analyzing the marker index and the recombinant of the phenotype-linked markers to identify and narrow the location region of the target genes. The experimental effect would be better if the number of plants were higher, however, this would increase the workload and need more planting space. Based on our experiment, it was found that the 30–100 plants used in this step would be appropriate to achieve better experimental effect with less workload and planting space. BC_2_F_1_ population also could be used here if cloning the target genes as soon as possible was desirable and there were enough markers. The markers (most were repeat sequence-based markers) were downloaded from the sharing platform on the internet or identified from the resequencing results in Step Three. The markers around the candidate region got in Step Three were the priority choices. Then, the allele polymorphisms of these markers among the individuals were detected by PCR (Polymerase Chain Reaction) and the markers linked to the phenotype characters were identified. For the recombinants of the markers located at different sides of the target gene were different and their marker indexes would be close to 0.5, the target gene could be narrowed among two markers (Figure 1).

Step Four: Identification of the mutated site. After the candidate region was found, the mutational site (the Δmarker-index is about 0.5) in the target region was identified by using the resequencing genome data when the sequence of the mutated gene was changed, and the mutated site was verified in the recombinants (Figure 1). When the mutants were caused by the changes of the epigenetic modifications around the target genes, the target region could be obtained and the candidate target genes could be identified by RNA-seq or RT-PCR (Real time-PCR).

For the homologous mutants, we crossed them with the closely related varieties to produce F_1_ generation without abortion, and then the F_2_ population was used as the mapping population (Appendix A). The following procedures were similar to the cloning method of the heterozygous mutants except that the marker indexes and Δmarker indexes near the mutated site were about 1.

To illustrate this method in detail, the cloning processes of one *Arabidopsis* heterozygous mutant and two rice homologous mutants, whose mutated sites were not obtained by the whole-genome sequencing method, are shown below.

### 2.2. Cloning Process of Arabidopsis Heterozygous Mutant 275-3

The mutant *275-3* was one of the embryo-defective mutants purchased from *Arabidopsis* Biological Resource Center (http://abrc.osu.edu/; accessed on 7 June 2021). During the vegetative developmental stage, the features of heterozygous mutant *275-3* were similar to WT (Col), but abnormal albino ovules in its silique were produced during the reproductive development stage (Figure 2a,b). The detailed data of the heterozygous mutant *275-3* exhibited that there were 26.49% albino ovules (*n* = 789) and 69.35% offspring (*n* = 62) producing albino ovules, implying the phenotype of *275-3* was controlled by a recessive gene (Table 1). To research the cause of the albino ovules, we cleared the *275-3* ovules at different developmental stages and found that the embryos of the albino ovules were stagnating in the globular stage compared to WT (Figure 2c–h). In the globular stage of the WT ovules, the embryos in the albino ovules were still below 16 cells, indicating that the embryo development was delayed (Figure 2c,f). As the WT ovules developed to form the torpedo embryos, the embryos of the albino ovule were still in the early globular stage, although its number of embryonic cells increased (Figure 2d,e,g,h). Meanwhile, the endosperm development in globular, heart and torpedo stages were also observed by detecting the endosperm auto-fluorescence. The results showed that the endosperm free nuclei of WT increased quickly from the globular stage to the torpedo stage, while the number of endosperm free nuclei was significantly less in *275-3*, indicating that the development of endosperm in *275-3* was slower than in WT (Figure 3). All of the above showed that the development of the embryo and the endosperm were both affected and the ovules were stagnating in the globular stage, leading to the abortion of seeds.

To clone the mutated gene, firstly, we crossed *275-3* with another ecotype L*er* (Landsberg *erecta*) to produce F_1_ generation. The mutant of the F_1_ generation was backcrossed with L*er* three times to get BC_2_F_1_ population and BC_3_F_1_ population. In the BC_1_F_1_ and BC_2_F_1_ generations, the abortion ratios of the mutant were 20.66% and 22.73% respectively, confirming that the mutant *275-3* was caused by a recessive gene (Table 1). Because there were enough markers between *Arabidopsis* ecotype Col and L*er*, and we wanted to cut down the time of gene cloning, Step Three was performed earlier using the 39 plants of BC_2_F_1_ population to identify and narrow the region where the target gene was located, through analysis of the phenotype-linked markers and their recombinants. We downloaded and synthesized the primers of SSLP (simple sequence length polymorphism) markers from TAIR database (The *Arabidopsis* Information Resource, https://www.arabidopsis.org/servlets/mapper; accessed on 7 June 2021). The marker indexes (the rate of Col-type gametophyte) of 25 SSLP markers were calculated. The results showed that only the indexes of markers in Chromosome 2 were more than 0.4 (Appendix A). Among them, the indexes of F3P11 and T20P8 both were 0.4872, implying that the mutated gene was around here. Two other markers, F27L4 and T19L18, were chosen to analyze the marker indexes with PCR technology, indicating that they were 0.4872 and 0.5, respectively. Besides, the PCR results showed that the 13th plant was the recombinant at T23K3, T12H3, F3P11 and F27L4, which was different from the recombinant (the 25th plant) at T16B24 and T20P8, telling us that the target region was about 1.53Mbp between F27L4 (Chr2: 10,068,751 bp) and T20P8 (Chr2: 11,595,846 bp) (Appendix A).

After the BC_3_F_1_ population formed, Step Two was performed and the DNA pool of 136 BC_3_F_1_ plants was resequenced with 40× depth through the second-generation sequencing technology as the mutant sample, while the DNA pool mixed with 20 L*er* plants was resequenced as the control. The sequencing data was mapped on the *Arabidopsis* reference genome (Col). The polymorphic loci (mainly SNP and InDel) were identified and their Δmarker indexes were calculated. All of these data were analyzed using MutMap method [7]. The average distributions of the Δmarker-index showed that there were 30 candidate regions with the threshold 0.45 (Figure 4a). The regions were too many to get the right one and there also were too many mutated sites in the most likely region (Chr2: 7~17 M). Thus, it was difficult to clone the target gene only using the MutMap method.

To clone the target gene, we used the genome sequencing data in the cloning procedure, and verified the SSLP primers downloaded from TAIR database with the resequencing data as described as Step Three. Five markers between F27L4 and T20P8 were selected to analyze their polymorphisms in the BC_3_F_1_ recombinants at F27L4 and T20P8. The results showed that the 58th plant was recombinant at T28I24, F27A10, and T22F11 while the 26th, 63rd and 65th plants were recombinants at F12C20 and T9J22 (Figure 4b). Thus, the mutated gene was in the 0.54Mbp region between T22F11 (Chr2: 10,779,026 bp) and T9J22 (Chr2: 11,328,569 bp). In this region (Chr2: 10,779,026 bp-11,328,569 bp), only one mutation was found in the exon after removing the sequence difference between Col and L*er*. A cytosine was inserted after Chr2: 11,104,201 bp leading to the shift of *At2g26060* ORF (open reading frame) (Figure 4c). Then, the DNA fragment containing this mutated site was cloned and sequenced. The results of four mutant plants showed that the mutant *275-3* was a heterozygote; one allele was the same as WT while the other had one extra cytosine (Figure 4d). In *Arabidopsis*, *At2g26060* encoded a homolog of the yeast cytosolic iron-sulfur protein assembly protein 1 (also known as *CIA1*). It was reported that AtCIA1 formed a complex with AtAE7, AtNAR1 and AtMET18 to promote the transfer of Fe-S clusters to the apoprotein located in the cytoplasm and nucleus, which was essential for maintaining the integrity of *Arabidopsis* genome [24]. The mutation of AtCIA1 produced albino ovules and stagnation embryo phenotypes, similar to the mutant *275-3* [24]. Therefore, we thought that the mutation of *AtCIA1* was the cause of the phenotypes of the heterozygous mutant *275-3* (Figure 4e).

### 2.3. Cloning Process of Rice Mutant ygs

Apart from the *Arabidopsis* mutant, we also used the improved method to clone rice homologous mutants that were not cloned by the whole-genome sequencing method. Rice mutant *ygs* (yellow-green seedling) without the insertion of T-DNA was obtained from the RISD DB (Rice T-DNA Insertion Sequence Database) and its background was cultivar Hwayoung (HY). The mutant *ygs* seeds germinated to produce yellowish-green seedlings, which faded as they grew (Figure 5a). Finally, the mutant *ygs* turned white and died. In order to determine whether the phenotype of the mutant *ygs* was controlled by a single gene, we calculated the phenotypic segregation ratio and found that the segregation ratio of the individuals derived from heterozygotes was 2.58:1 (green: yellowish-green = 62: 24), which was consistent with the expected genetic segregation ratio 3:1 (*p* = 0.53, Chi square test), indicating that the mutant *ygs* was a recessive trait controlled by a single gene in HY. To clone the *YGS* gene, heterozygote *YGS*/*ygs* was crossed with the *japonica* cultivar Zhonghua 11 to get the F_2_ population as described by the Step One in Appendix A. Contrary to expectations, the ratio of the homologous mutant *ygs* in the progenies derived from the cross (Zhonghua 11♀× *ygs*♂) was 8.1% (3/37) that was close to 1/16, implied that *ygs* in F_2_ population might be controlled by two recessive genes. The result was different from the segregation ratio in HY, therefore, we proposed that one gene, essential for the *ygs* phenotype, might be mutated in the cultivar HY but not in the cultivar Zhonghua 11.

Then, we performed Step Two of the SBM method to sequence the *ygs* pool mixed with equal DNA of 32 *ygs* plants and the HY pool mixed with equal DNA of 30 HY plants using the Illumina sequencing platform. The sequencing data of the *ygs* pool was 40× depth, while the HY pool was 20×. The clean data was screened out and mapped on the rice reference genome and the RGAP 7.0 gene modes (http://rice.plantbiology.msu.edu/pub/data/Eukaryotic_Projects/o_sativa/annotation_dbs/pseudomolecules/version_7.0/; accessed on 7 June 2021). We identified the variants SNP and InDel, and excluded the markers whose supported read numbers were not more than eight. Then, we performed two analysis methods to identify the candidate sites. Firstly, we calculated the G value of SNP and drew the Manhattan plot to identify the phenotype-linked markers (Figure 5b). The top 0.1% sites were considered as the candidates, and 83 possible sites appeared, five of which were the most likely sites (Figure 5b, Appendix A). Meanwhile, we calculated the Δmarker indexes of the SNP and InDel markers and analyzed them using MutMap [7]. The average distributions of the Δmarker-index showed that there were eight candidate regions in seven chromosomes with the threshold 0.6 (Figure 5c).

However, the candidate sites were too many and further analysis found that the target gene was not in these candidate regions obtained by the above methods, hence we conducted Step Three of the sequencing-based mapping method. The primers of SSR (simple sequence repeat) markers were downloaded from Gramene database (https://archive.gramene.org/markers/microsat/; accessed on 7 June 2021) and verified by the resequencing data. The 36 markers were chosen for PCR to analyze their marker indexes in the *ygs* mutant, and the results of seven plants showed that only one possible marker RM1080 (Chr12: 906,291 bp) was linked to the target site for its marker-index was 0.9286, which was close to 1 (Appendix A). Besides RM1080, we also screened out another two markers to clone this target gene in 25 *ygs* mutants. The recombinants at RM1880 (Chr12: 747,262 bp) and at RM1080 both were the 4th and 14th plants, while there were seven recombinants at RM27618 (Chr12: 3,804,685 bp) (Figure 6a). For the 4th plant, there were recombinants at the three markers, we thought that three markers were located in the same side of the target gene. To get the target gene, we analyzed the SNP and InDel markers around this region (Chr12: 1 bp-747,261 bp) and removed the markers whose Δmarker indexes were lower than 0.5. As the Δmarker indexes around the target gene should be 1, the target region was narrowed in the region (Chr12: 80,442 bp-747,261 bp) (Figure 6b). Using IGV (Integrative Genomics Viewer) software, we screened out variants in this region in the sample *ygs*, HY and ZH11 (Zhonghua 11, which was sequenced in another project), and found that there was a 23.5Kb gap in *ygs* but not in HY and ZH11, indicating that the gap might be a mutation in *ygs* but not caused by sequencing technology (Figure 6c). The deletion of the 23.5 Kb sequence led to the mutation of four genes, and one of them, LOC_Os12g01210, encoded a pentatricopeptide repeat-containing protein. This gene has been reported by Mao et al. and has functional redundancy with LOC_Os11g01210. It was reported that the simultaneous mutation of LOC_Os12g01210 and LOC_Os11g01210 produced an etiolated seedling and could not form mature plants [25], which was similar to our observed phenotypic feature in *ygs*. To verify that the phenotype of the *ygs* mutant was caused by the mutation of LOC_Os12g01210 and LOC_Os11g01210, we observed the sequencing data of LOC_Os11g01210 in *ygs*, HY and Zhonghua 11, and found that there was a ~300 bp deletion in HY but not in Zhonghua 11 (Figure 6d). Thus, we thought that the phenotype of the *ygs* mutant was caused by the double mutation of LOC_Os12g01210 (OsYGS1) and LOC_Os11g01210 (OsYGS2). 

### 2.4. Cloning Process of Rice Mutant abs

Meanwhile, another rice mutant was also obtained from the RISD DB (Rice T-DNA Insertion Sequence Database) and its background was cultivar HY. The new mutant germinated normally and grew like WT for about ten days. Then, the abnormality of the mutant appeared: the green of the third leaf faded and the newly emerged fourth leaf was white and chlorotic (Figure 7a,b). After that, the growth of the mutant was arrested and the seedlings died gradually, Thus, this mutant was named as *abs* (abnormal seedling). To clone the mutated site, we crossed *abs* with ZH11 to get the mapping F_2_ population, and calculated the segregation ratio in the F_2_ population. In 227 plants, there were 50 mutants which exhibited a 3.54:1 segregation ratio and was consistent with the expected genetic segregation ratio 3:1 (*p* = 0.30, Chi square test). This implied that the *abs* mutant was controlled by a recessive gene.

Then, the *abs* DNA pool of 50 plants was resequenced on the Illumina sequencing platform with 40× depth as described in Step Two. The clean data was screened out and mapped on the rice reference genome and the RGAP 7.0 gene modes (http://rice.plantbiology.msu.edu/pub/data/Eukaryotic_Projects/o_sativa/annotation_dbs/pseudomolecules/version_7.0/; accessed on 7 June 2021. The data was analyzed with the resequencing data of Hwayoung and Zhonghua 11 to identify the variants (SNP, InDel, SV and CNV). The Δmarker indexes of SNP were analyzed using the MutMap method and the average distributions of the Δmarker-index showed that there were 14 candidate regions in seven chromosomes with the threshold 0.5 (Appendix A).

Then, we conducted Step Three to identify the markers linked to the mutated site using the SSR primers used in the cloning of *OsYGS1/2*. The results of 30 plants showed that the marker indexes of two markers, RM1080 and RM1880, were more than 0.7, implying that the mutated site was linked to the two markers but not to the candidate regions identified by the MutMap method (Appendix A). As the recombinants at RM1080 and RM1880 were the same and most were different from the recombinants at RM519, we concluded that the mutated site of *abs* was located between RM1080 and RM519 (Appendix A). In this region, another five SSR markers were screened out using the resequencing data and the Gramene database (https://archive.gramene.org/markers/microsat/; accessed on 7 June 2021). The PCR results showed that the recombinants at RM27618 were the 6th, 8th, 15th, 19th and 26th plant which were different from the recombinants at RM27715 (the 1st, 3rd, 5th and 29th plant), and there was no recombinant at RM27694 and RM27697 (Appendix A). Thus, the mutational site should be in the 1.32M region between RM27618 (Chr12: 3,804,685 bp) and RM27715 (Chr12: 5,189,063 bp).

This region was still too long with too many candidate variants, so we chose another 4 InDel markers from the resequencing data and another 106 mutants of the F_2_ population to narrow the target region. Exclude the false-positive plants, 14 recombinants at RM27618 or RM27715 were found and used for later research (Appendix A). The PCR results showed that there were the same recombinants (17th, 24th, 42nd, 66th, 84th, 94th and 104th) at InDel 2, InDel 3 and InDel 4, which were different from the recombinants at RM27694 and RM27697 (14th and 64th) (Figure 7d). These indicated that the mutational site was between InDel 4 (Chr12: 4,403,614 bp) and RM27694 (Chr12: 4,814,750 bp). In this region, we screened the variants using the resequencing data and found that there were two variants and only one located in the exon. With the help of IGV software, we found that a thymine (Chr12: 4,710,533 bp) was converted to adenine, leading to the conversion of Ile403 to Asn403 in LOC_Os12g09000. (Figure 7e,f). Then, the genetic co-separation experiment verified that this site was tightly linked to the phenotype of *abs* and the transgene of LOC_Os12g09000 genomic sequence complemented the phenotype of *abs* (Figure 7c). Thus, LOC_Os12g09000 was the target gene and this gene was essential for the survival of rice seedlings.

## 3. Discussion

In the study of genes, the forward genetic approach is a powerful tool for identifying genes, based on phenotypes of interest, which cannot be done by the reverse genetic approach. Unfortunately, the forward genetic approach is limited by the fact that it is very difficult to map the causative mutational sites. The most popular methods are the conventional map-based cloning method and the whole-genome sequencing method, including SHOREmap and MutMap [4,5,6,7]. The advantages of the conventional map-based cloning method are that it can clone all the types of mutation, including the deletion or insertion of several Kb of DNA fragments and the mutation of epigenetic modifications, and can get the area containing the target gene exactly (Table 2). Its downsides are that it is lengthy, requires a large mapping population and confirmation of many candidate genes in a broad genetic region to characterize the causal variant (Table 2) [8,9,10,11,12,13,14,15,16]. The whole-genome sequencing method accelerates the identification of causal mutations by defining the mapping regions and identifying genetic variants simultaneously. However, it could not be used to clone the mutation of epigenetic modification and to clone the variants with a certain failure probability for multiple reasons, especially for heterozygotes (Table 2) [26,27,28,29]. Only one of six mutants was cloned successfully and no mutated genes of heterozygous mutants were found in our assay. Thus, we applied the highly complementary characteristics of the two methods and developed the sequencing-based mapping method to clone genes that were not obtained by the whole-genome sequencing method in the diploid species with known genomic information.

Through the use of this new mapping method, we cloned six genes in five mutants of rice and *Arabidopsis*, and cited three examples in this report. The cloning process of *Arabidopsis* heterozygous mutant *275-3* showed that the whole-genome sequencing method obtained many candidate regions and a very broad region for the background noise, but failed to get the target gene. With the help of recombinant analysis, we found the right candidate region and narrowed the target region. Finally, with the help of the whole-genome sequencing data, the mutated site was obtained. In the cloning of rice mutant *ygs* and *abs*, we identified many markers and found the mutated sites after identifying the target regions by using SBM. In these process, 32 F_2_ plants for *ygs*, 136 F_2_ plants for *abs* and 136 BC_3_F_1_ plants for *275-3* were used for the fine mapping, while the numbers of plants used in the conventional map-based cloning method needed to be more than 600 plants [8,9,10,11,12,16,30]. So, this method just needed a small greenhouse to plant the mapping population, this is very user-friendly for most popular laboratories. The GWAS analysis (genome wide association study) also can achieve this goal by resequencing more than 10 pools (some are more than 100 pools) of plants and has the advantage of identifying many genes underlying the same phenotype or different phenotypes at the same time, especially in the research of a complex quantitative trait. However, it is more expensive which is a heavy burden for the popular laboratories (Table 2) [21,31]. Moreover, the SBM could also be used to clone the mutation of epigenetic modifications. We used it to get one candidate region in an embryo-defective mutant. In this region, no sequence changes had been found, and the two markers beside it were linked to the phenotype. We planned to use the RNA-seq method to identify the target gene and expected to get it. Thus, the SBM could clone different types of plant mutations effectively, and it is a powerful tool for the study of the functions of plant genes in diploid species with known genomic information.

In the polyploid species, the ratio of the homozygous genotypes in hybrid progenies is very low in the F_2_ population (1/36 in the tetraploid species) and there are more than three kinds (tetraploid species) of genotypes of heterozygotes, which makes the calculation of the allele-ratio index (marker index in Step Three) very difficult. It was found that the ratios of the homozygous genotypes at the phenotype-unlinked markers should be less than or about 1/36 (tetraploid species), while the ratios of the homozygous genotypes at the phenotype-linked markers are closer to 1.0. Therefore, if the SBM is used in the polyploid species, the ratio of the homozygous genotypes should be used as the marker-index. In order to identify the high-quality markers effectively in Step Two, both of the parents should be resequenced. Theoretically, the number of plants needed and the expected depth of sequencing are similar to that in the diploid species. Because the heterozygous mutants of the polyploid species contain several kinds of genotypes, it is difficult to clone them by this method and GWAS is the better choice. Thus, the SBM can be used to clone the homologous mutants of the polyploid species with some modification, as long as they have abundant genomic information. Because most of the crops have been de novo sequenced and their genomic information is abundant, this method will benefit the research of the crops. In the species without abundant genomic sequences, this method may be restricted and the de novo sequencing should be done first.

Besides, because of hybrid sterility, the F_1_ and F_2_ generations hybridized between many rice varieties have a certain abortion rate, which interferes with the separation of rice heterozygous mutants from the wild type in F_2_ generation plants. Therefore, heterozygous mutants of rice can only be hybridized with the varieties closely related to the mutants. However, it is difficult to screen enough molecular markers to clone the target genes using the conventional map-based cloning method. Most of these mutated genes in rice heterozygous mutants are essential for the formation or development of rice seeds. Therefore, the SBM can be used to clone many heterozygous genes, to study the phenotypic characteristics of heterozygous mutants, and to dissect the molecular regulated mechanisms in the formation and development of rice seeds, which would provide a firm foundation for the improvement of rice varieties.

## 4. Materials and Methods

### 4.1. Plant Materials

*Arabidopsis thaliana* ecotype Columbia (Col), Landsberg *erecta* (L*er*) and the mutant *275-3* (the background was Col) were grown in the greenhouse of Wuhan University at 22 ± 2 °C under a 16 h light/8 h dark photoperiod. The mutant *275-3* was obtained from the *Arabidopsis* Biological Resource Center (http://abrc.osu.edu/; accessed on 7 June 2021).

The rice mutant *ygs* (*yellow-green seedling*) and *abs* (*abnormal seedling*), obtained from RISD DB (Rice T-DNA Insertion Sequence Database), wild-type varieties (*Oryza sativa* L. *ssp*. *Japonica*, cultivar ‘Hwayoung’, HY; cultivar ‘Zhonghua11′, ZH11) were grown in natural environment conditions or in the greenhouse at 28 ± 2 °C under a 14 h light/10 h dark photoperiod at Wuhan University. Plant phenotypic features were photographed using a digital camera (Nicon D5000 and Micro NIKKOR 60 mm, Japan).

### 4.2. Ovule Clearing

Fresh ovules of Col and the mutant *275-3* were isolated from siliques using forceps and soaked in the Hoyer’s solution (chloral hydrate: glycerol: water= 8:1:2 (*w*/*v*/*v*)) for 5-–30 min (ovules at globular stage) or 1–2 h (ovules at heart and torpedo stages) [32]. Then, the cleared ovules were observed and photographed with differential interference contrast microscopy (Olympus TH4-200 equipped with a CCD of a SPOT digital microscope camera).

### 4.3. Observation of Endosperm Phenotype

The fresh siliques were dissected and transferred into the fixative (4% glutaraldehyde in PBS, pH 7.0), vacuumed until all siliques were sunk in the fixative, and fixed overnight at room temperature after siliques were placed into fresh fixative. Next, the samples were dehydrated by a series of graded alcohols (15%, 30%, 50%, 70%, 90% and 100%) and rehydrated by graded alcohols (90%, 70%, 50%, 30%, 15% and 0%) for 20 min for each gradient, Then, the ovules were separated and mounted onto the slides with Hoyer’s solution until the samples were cleared. Finally, the embryos and endosperms in transparent ovules were observed using a confocal laser scanning microscope (Olympus FluoView FV1000) with 488 nm excitation [33].

### 4.4. DNA Extraction

A suitable amount of fresh leaves (about 0.2 g) of *Arabidopsis* thaliana and rice were ground and crushed with the help of liquid nitrogen. Then, the powders were respectively digested in 700 mL 65 °C CTAB extraction solution (2% CTAB, 2% PVP-40, 1.42 M NaCl, 20 mM EDTA, 0.2% *β*-mercaptoethanol, 10 mM Tris-HCl) for half an hour, and centrifuged at 12,000 rpm for 5 min to get the supernatant. The reagent 570 mL chloroform: isoamyl alcohol (volume ratio: 24:1) was added in the supernatant, shaken violently, and centrifuged at 12,000 rpm for 10 min to purify their DNA (the supernatant). The supernatant was purified again using 570 mL chloroform: isoamyl alcohol (volume ratio: 24:1) and then was transferred into a new EP tube, equal volume of isopropanol was added, and stored at −20 °C for 30 min. Then, the mixture was centrifuged at 12,000 rpm for 5 min, and the supernatant was discarded and 500 mL 75% alcohol was added to remove the residual isopropanol. Finally, the mixture was centrifuged at 12,000 rpm for 2 min and put in 37 °C for 10 min to remove alcohol, and 50 μL of ddH_2_O containing RNase was added to dissolve the DNA. The DNA extracted by this method had high purity and could be used for genome resequencing and sequencing-based mapping.

### 4.5. Whole-Genome Sequencing and Data Analysis

The heterozygous mutant *275-3* was crossed with L*er* to obtain F_1_ generation that exhibited the same phenotypes as *275-3*, and the F_1_ generation was backcrossed with L*er* three times to get the BC_3_F_1_ population that was used for the whole-genome sequencing. Meanwhile, rice homologous mutants were crossed with ZH11 to get the F_2_ population that was used for the whole-genome sequencing. The L*er* and ZH11 were also resequenced as the control, and the leaves from the sequencing population were collected to extract DNA as described above. The DNA quality of the samples was detected by the ultraviolet spectrophotometer NanoDrop2000 (Thermo, USA) to ensure that OD_260_/OD_280_ ≥ 1.8, OD_260_/OD_230_ ≥ 2.0, and DNA concentration ≥ 100 ng/μL. The 1% agarose gel electrophoresis was used to ensure that the genome did not degrade. Then, equal amounts of DNA for every sample (more than 30 samples for mutants and 20 samples for WT) were mixed (5–10 μg) and wer resequenced using the second-generation sequencing technology on the Illumina sequencing platform in Wuhan Seqhealth Co., Ltd. except for sample ZH11 that was sequenced in the Shanghai Oebiotech Corporation.

Using Trimmomatic software, the raw data obtained by the Illumina sequencing platform was analyzed to obtain high-quality sequencing reads (the clean reads, more than 95% of whose nucleotide sequences had Q-score of 30) by removing reads containing the adapter, reads containing ploy-*n* and low-quality reads. Then, the clean reads of L*er* and the mutant *275-3* were mapped on the *Arabidopsis* reference genome (https://www.arabidopsis.org/download/index-auto.jsp?dir=%2Fdownload_files%2FGenes%2FTAIR10_genome_release; accessed on 7 June 2021), while the clean reads of HY, ZH11 and rice mutants were mapped on the rice reference genome (MSU Rice Genome Annotation Release 7, http://rice.plantbiology.msu.edu/pub/data/Eukaryotic_Projects/o_sativa/annotation_dbs/pseudomolecules/version_7.0/all.dir/all.con; accessed on 7 June 2021) using BWA software. Using GATK software, SNP (single nucleotide polymorphism), InDel, structural variations (SV, mainly insertion and deletion of large fragments), and copy number variations (CNV) of the experimental samples were identified, and their numbers in different pools were calculated. Then, the marker indexes (allele-ratio) of SNP and InDel markers were calculated using the formula: marker-index = alt-number/(alt-number + ref-number) (alt-number indicated the number of the reads that contained the allele different from the reference genome, ref-number indicated the number of the reads that contained the same allele of the reference genome), and the markers with more than 8 supported-reads were used for the later analysis. Using the MutMap method described by Abe, the Δindexes of all markers were calculated and analyzed [7]. The average distributions of the Δmarker indexes were calculated and plotted by using the sliding window approach with a 100 Kb window size and 20 kb sliding step except rice mutant *abs* which used the sliding window approach with a 500 Kb window size and 100 kb sliding step. Meanwhile, the G value was calculated using the formula $G=2\sum_{i=1}^{4}n_{i}\ln\left(\frac{n_{i}}{\hat{n_{i}}}\right)$, $\hat{n_{i}}=\left(n_{1}\times n_{2}\right)\left(n_{1}\times n_{3}\right)÷\left(n_{1}\times n_{2}\times n_{3}\times n_{4}\right)$, and *n*_1_, *n*_2_, *n*_3_ and *n*_4_ were the supported read numbers of the two alleles in two samples. The top 0.1% of G values were considered as the sites significantly linked to the phenotype of rice mutant *ygs* [34].

### 4.6. Identification and Analysis of Mapping Markers

To clone the mutated gene in *Arabidopsis* mutant *275-3*, the primers of SSLP (simple sequence length polymorphism) markers were downloaded from TAIR database (The *Arabidopsis* Information Resource, https://www.arabidopsis.org/servlets/mapper; accessed on 7 June 2021). Some SSLP markers were verified by the whole-genome sequencing data and were chosen for mapping if there was an insertion or a deletion of more than 5 bp at the sites (for the size difference of more than 5bp could be separated by the 4% agarose electrophoresis). Some primers of rice SSR (simple sequence repeat) were downloaded from Gramene database (http://www.gramene.org/; accessed on 7 June 2021), and confirmed by the whole-genome sequencing data with the same threshold in *Arabidopsis*. The others were designed by the software Primer premier 6 around the InDel markers (with insertion or deletion of more than 5 bp) identified from the whole-genome sequencing data. The PCR (Polymerase Chain Reaction) products of the mapping population were detected by the 4% agarose electrophoresis and the marker indexes (the rate of the Col genotypes or HY genotypes) were calculated to identify the markers that linked to the mutational site. Furthermore, the distributions of recombinants at these phenotype-linked markers were analyzed to narrow the candidate region, as the number of recombinants should be less when the marker was closer to the mutational site and the recombinants at the markers on different sides of the mutated site should be different. Finally, using the resequencing genome data, the mutational site in the target region was identified.

### 4.7. Plasmids Construction and Genetic Transformation

To verify the function of *OsABS* (*LOC_Os12g09000*), the genomic sequence of *LOC_Os12g09000* was cloned and inserted into the *pCAMBIA1300* vectors. The vector was introduced into *Agrobacterium tumefaciens* strain EHA105 and transformed into callus of rice mutant *abs* [35].

### 4.8. Primers

The primers used in this study are listed in Appendix A.

## Figures and Tables

**Figure 1 ijms-22-06224-f001:**
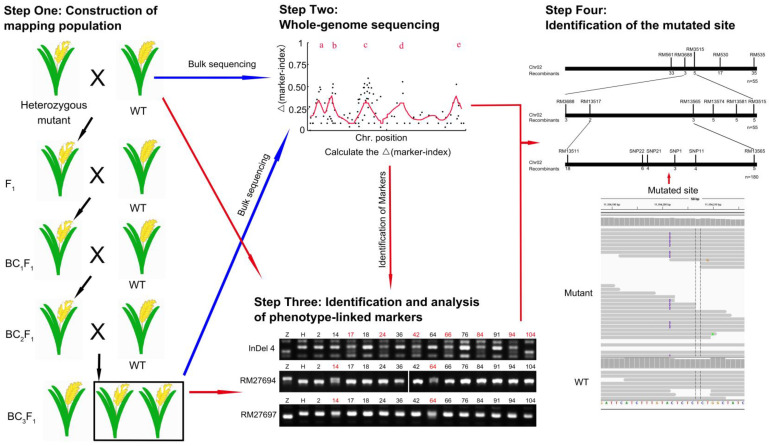
The simplified cloning scheme for the heterozygous mutant. Step One: the heterozygous mutant was crossed with WT of the other variety four times to generate the BC_3_F_1_ population. Step Two: about 30–200 heterozygous plants of BC_3_F_1_ population were selected to extract DNA, and the DNA samples were mixed with equimolar of each, resequenced, and analyzed. The markers (mainly SNP and InDel) were identified and their marker indexes were calculated to identify the high-quality markers for identifying the candidate regions using the MutMap method. Step Three: the BC_3_F_1_ population was used for the analysis of the recombinants at the phenotype-linked markers to identify and narrow the candidate region with the markers which were downloaded from the sharing platform on the internet or identified from the resequencing results. Step Four: using the resequencing genome data, the mutational sites (the Δmarker-index was about 0.5) in the target region were identified.

**Figure 2 ijms-22-06224-f002:**
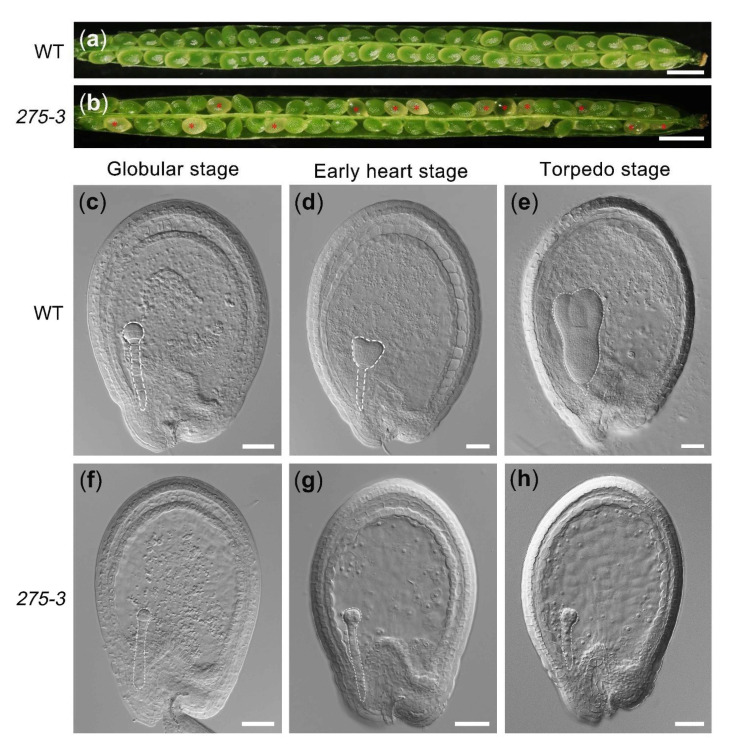
The phenotype of *Arabidopsis* mutant *275-3*. (**a**) The silique of WT. (**b**) The silique of *Arabidopsis* mutant *275-3;* the red asterisks indicated the albino ovules. (**c**–**e**) The WT ovules at the globular stage (**c**), early heart stage (**d**) and the torpedo stage (**e**). (**f**–**h**). The ovules in the mutant *275-3* at which WT ovules were at the globular stage (**f**), early heart stage (**g**) and the torpedo stage (**h**). The bars were 1 mm in (**a**) and (**b**), and 50 μm in (**c**–**h**).

**Figure 3 ijms-22-06224-f003:**
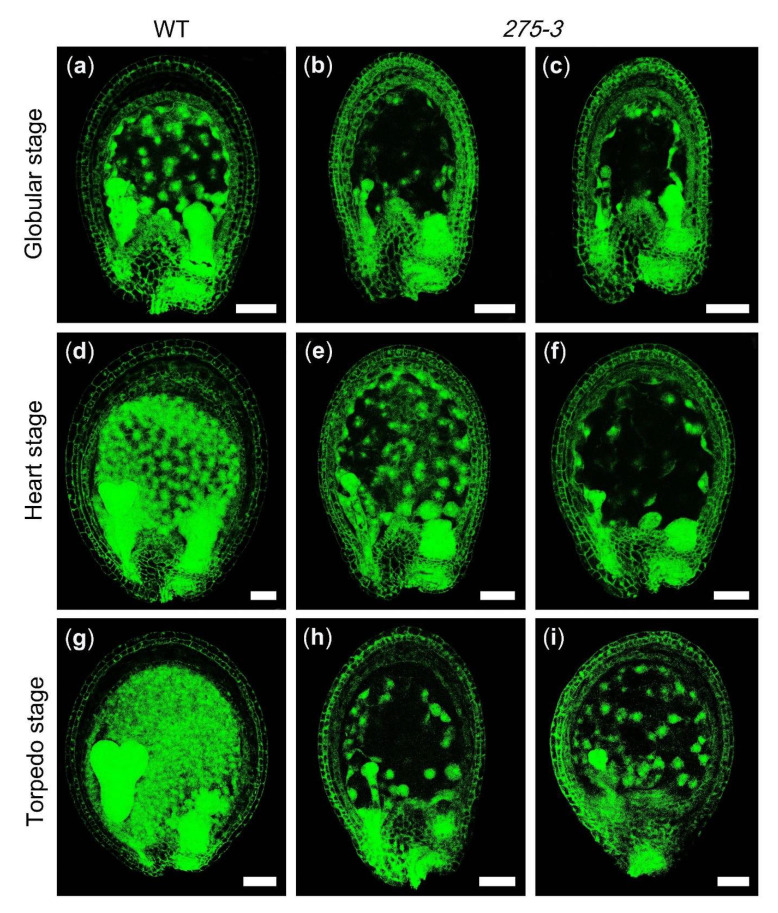
The developmental process of endosperm in *Arabidopsis* mutant *275-3*. (**a**,**d**,**g**) WT ovules. (**b**,**c**,**e**,**f**,**h**,**i**) The ovules in the mutant *275-3*. Scale bars = 50 μm.

**Figure 4 ijms-22-06224-f004:**
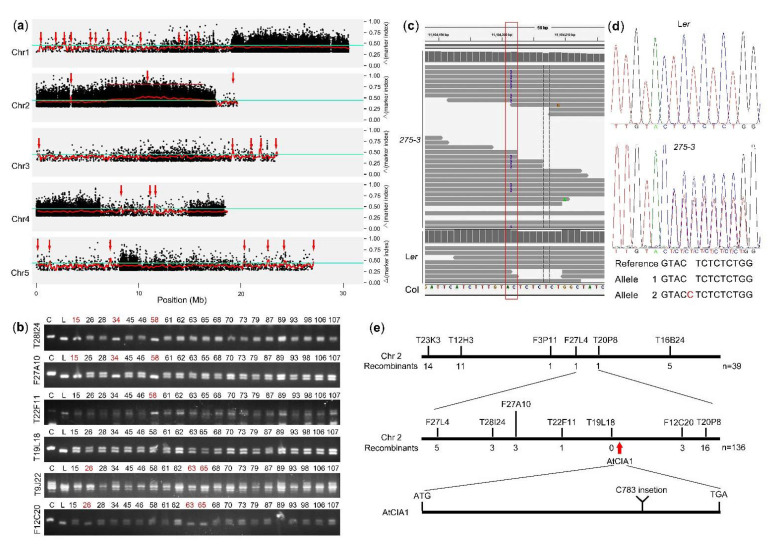
The cloning data of *CIA1* in the mutant *275-3* of *Arabidopsis*. (**a**) The Δmarker-index plots of the mutant *275-3* generated by the analysis of MutMap. The red lines represent the sliding window average Δmarker-index values of the 100Kb interval with 20kb increments. The blue lines indicated the threshold 0.45 to identify the candidate regions. (**b**) The PCR results of some SSLP markers in the fine mapping of *275-3.* The plants of recombinants at F27L4 or T20P8 were used to analyze the recombinants at some markers (at the left side) and the red numbers indicate the recombinants. T28I24, F27A10, T22F11, T19L18, T9J22 and F12C20 are located around Chr2: 10,382,120 bp, Chr2: 10,516,283 bp, Chr2: 10,779,026 bp, Chr2: 11,061,880 bp, Chr2: 11,328,569 bp and Chr2: 11,462,271 bp, respectively. C, Col; L, L*er*. (**c**) The whole-genome sequencing results at the mutated site in *275-3* and WT using the IGV software. (**d**) The sequencing results of DNA fragments containing the mutated site in WT and *275-3*. (**e**) The cloning procedure of *CIA1* in *Arabidopsis*.

**Figure 5 ijms-22-06224-f005:**
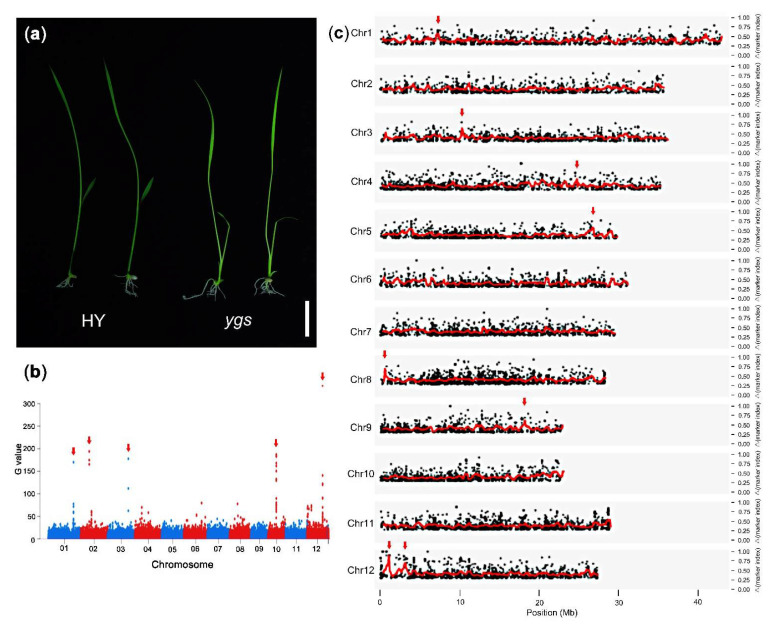
The phenotype of rice mutant *ygs* and the cloning data using the whole-genome sequencing method. (**a**) The phenotype of 7DAG seedlings in rice mutant *ygs.* Bar was 25 mm. (**b**) The Manhattan plot of the G values. The red arrows indicated the 5 most likely candidate sites. (**c**) The Δmarker-index plots of the mutant *ygs* generated by the analysis of MutMap. The red lines represent the sliding window average Δmarker-index values of the 100 Kb interval with 20 kb increments. The red arrows indicated the most likely candidate sites.

**Figure 6 ijms-22-06224-f006:**
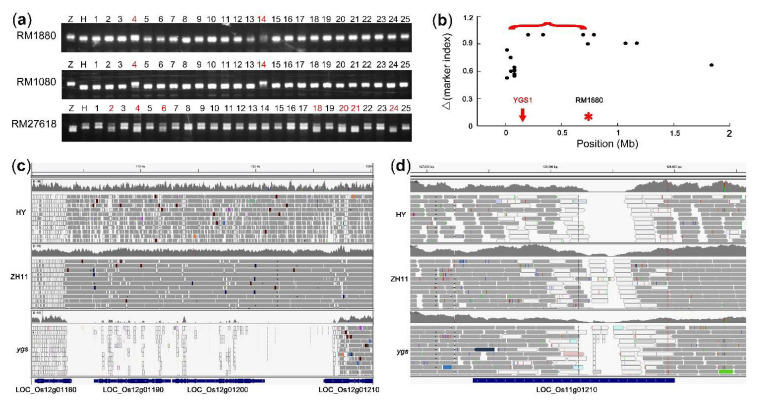
The cloning data and information using the sequencing-based mapping method. (**a**) The PCR results of three SSR markers in the fine mapping of *ygs.* The markers are shown at the left side and the red numbers indicated the recombinants. Z, Zhonghua 11; H, Hwayoung; RM1880, RM1080 and RM27618 are located around Chr12: 747,262 bp, Chr12: 906,291 bp and Chr12: 3,804,685 bp respectively. (**b**) The cloning data of OsYGS1, the red brace indicates the candidate region and the red arrow indicates the site of OsYGS1. (**c**) The whole-genome sequencing data at the site of OsYGS1 (LOC_Os12g01210) viewed by IGV software. (**d**) The whole-genome sequencing data at the site of OsYGS2 (LOC_Os11g01210) viewed by IGV software.

**Figure 7 ijms-22-06224-f007:**
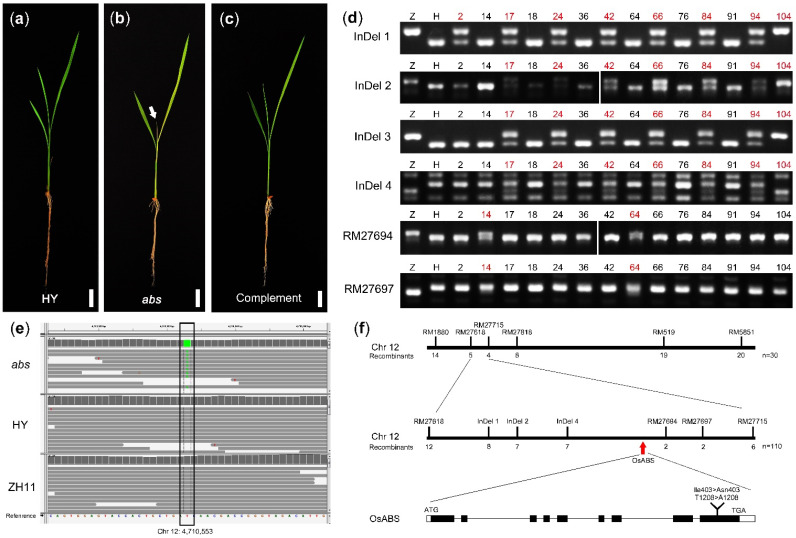
The phenotype of *abs* and cloning of OsABS gene: (**a**) 14 DAG seedling of Hwayoung (HY); (**b**) 14 DAG seedling of *abs*. The white arrow indicated the chlorotic fourth leaf; (**c**) 14 DAG complementary seedling of *abs*. (**d**) The analysis of recombinants around the mutated site in *abs*. All the plants used here were the recombinants at RM27618 or RM27715, and the red number indicated the recombinant. InDel 1, InDel 2, InDel 3, InDel 4, RM27694 and RM27697 are located around Chr12: 4,070,394 bp, Chr12: 4,187,138 bp, Chr12: 4,400,174 bp, Chr12: 4,403,614 bp, Chr12: 4,814,750 bp and Chr12: 4,920,770 bp, respectively. Z, Zhonghua 11; H, Hwayoung. (**e**) The whole-genome sequencing results at the mutated site in *abs,* HY and ZH11 using the IGV software. A thymine (Chr12: 4,710,533 bp, indicated in the black box) was converted to adenine. (**f**) The cloning procedure of *OsABS.* Bars were 2 cm in (**a**–**c**).

**Table 1 ijms-22-06224-t001:** The abortion ratios of *Arabidopsis* mutant *275-3* and its progenies.

Plants	Number of the Abortive Ovules	Number of the Normal Ovules	Number of the Total Ovules	Abortion Rate (%)
*275-3*	209	580	789	26.49
BC_1_F_1_	275	1331	1173	20.66
BC_2_F_1_	235	799	1034	22.73
Col-0	8	1199	1207	0.66
L*er*	2	1116	1118	0.18

**Table 2 ijms-22-06224-t002:** Comparison of different cloning methods.

Methods	Mutation Types	Time ^a^	Number of Population Plants	Planting Room	Number of Pools ^b^	Cloning Efficiency ^c^
Conserved map-based cloning	All	3–5 years	>600	large	0	++
Sequencing-based mapping (SBM)	All	1–2 years	30–200	small	2	+++
Whole-genome sequencing	Homologous SNP and InDel	170–390 days	20–200	small	2	+
GWAS	All	>170 days	>200	large	>10	+++

^a^: The costing time was calculated in rice, based on the following calculation criteria: the time for rice to grow one generation was 110 days, the time for resequencing and data processing was 30 days, the time for the verification of the mutated sites was 30 days, and all the procedures were thought to be performed well. ^b^: The pools here were thought to be resequenced. ^c^: Cloning efficiency represented the success rate for each method. + represented low success rate; ++ represented medium success rate; +++ represented high success rate.

## Data Availability

The supporting data was included in the paper and the Appendix A.

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
