# Peer review of "The Sequencing-Based Mapping Method for Effectively Cloning Plant Mutated Genes"

_ijms, 2021, doi:10.3390/ijms22126224_

Round 1

Reviewer 1 Report

Dear editor and colleagues, I have concluded my evaluation of the resubmitted manuscript “An improved map-based cloning method for effectively cloning plant mutated genes”. It is a study focusing on a combination of experimental and in silico procedures in order to map populations and identify mutants causing phenotypic differences, using a minimal number of plants. The authors have provided an implementation of their method and successfully identified mutants for Arabidopsis and rice. Despite possible limitations for mapping plants where genomic data are lacking, several researchers could benefit from this manuscript; hence I believe that it has merit for publication in the IJMS The authors have addressed all previously raised comments on my behalf and have provided a more grammatically sound manuscript As a result, my recommendation is acceptance of the manuscript

Author Response

Dear reviewer:

Thanks very much for your approval.

Reviewer 2 Report

The manuscript by Yu et al. reports an improved map-based cloning method useful for cloning plant mutated genes. As the authors raised, although the conventional map-based cloning method is indeed powerful to identify genes underlying phenotypes of interest, it is time-consuming because it requires a large mapping population and confirmation of many candidate genes in a broad genetic region to clone the causal variant. In this report, they proposed a new method (called an improved map-based cloning method, but I felt this name is not unclear, see below) which combines the conventional map-based cloning method with the whole-genome sequencing data of two pooled populations to screen out enough markers, and successfully demonstrated its effectiveness by using several examples of the plant mutants, Arabidopsis and rice. In particular, while this method has succeeded in identifying the causative gene with only about 32 ~ 136 plant mutants, the conventional method would require the use of 600 or more plants, suggesting that this method will significantly save time and costs. The manuscript is well written, and clearly showed the progress from the conventional methods, which would be of interest to many researchers in the related plant biology fields. Therefore, I can recommend publishing this work after addressing the following concerns:

Major points:

  1. Only the protocol of the new method is shown in Figure 1. However, to clarify the difference between the conventional and their improved method, it is better to show the schematic illustration of the conventional one corresponding to Figure 1. It would be helpful for non-expert for the Forward genetic approach.
  2. In addition, the difference between the names “conventional map-based method” and “improved map-based method” is not clear. If possible, please describe each method with a more descriptive name.
  3. In the conclusion part, it is stated that this improved method would be less costly than GWAS analysis. To clarify it more, it is better to compare the estimated costs of each method. In addition, it would be even better to compare the actual number of days it takes for each method (conventional-, and improved map-based approach, and GWAS analyses).

Minor points

  1. In Figure 5b, the red arrows could be found only 5 (not 7). Please confirm it carefully.
  2. In Figure 6cd, would OsYGS1 and OsYGS2 be correspond to Os12g01200 and Os11g01210, respectively? The figure and its legend do not correspond, and it is better to assign it more carefully.

Author Response

This manuscript is a resubmission of an earlier submission. The following is a list of the peer review reports and author responses from that submission.

Round 1

Reviewer 1 Report

Dear editor,

I'have evaluated the manuscript entitled "An improved map-based cloning method for effectively cloning plant mutated genes" . I believe that the topic which is about the cloning of mutant genes could be of interest for most IJMS readers. However, it lacks a detailed description of methods, there is no clear indications of the rational behind technical choices and therefore it is difficult to figure out if the proposed method could have a wide applicability. These, on my opinion, are major flaws for a methodological paper.

Moreover, the structure of the paper is inadequate and, in the present form, it is very difficult to follow the whole matter. 

Reviewer 2 Report

Dear editor and colleagues,

I have completed my review of the entitled manuscript “An improved map-based cloning method for effectively cloning plant mutated genes”. 
It is a research article that illustrates a ‘shortcut’ for mapping combining NGS and classical crossing procedures using two model plants (Arabidopsis and rice). Moreover, this procedure is applied for the mapping of both heterozygous and homozygous mutant genes
It is my opinion that the subject of the study is within the scope of the International Journal of Molecular Sciences and therefore has merit for publication.
The experimental processes seem robust and well executed and a functional complementation of putative target gene has been performed. I have however some remarks in order to make the manuscript more comprehensive.

More specifically, 

English proofreading is advised since there are several instances of grammar and syntax mistakes.

The authors focused only on model plants, hence the ‘proof of concept’ regarding crops were abundant genomic information is scare, cannot be applied easily. The authors must at least comment on that.

Moreover, Arabidopsis has one of the lowest c-values in terms of plant haploid genome content (circa 0.20 pg). Rice has also one of the smallest genomic size across small grains (circa 0.50 pg). The feasibility of the method in polyploids (autopolyploids and allopolyploids; wheat, oats etc) and the needed number of plants and/or the expected depth of sequencing has not been evaluated nor commented. Therefore, several caveats could arise when dealing with complex crops that also suppress or exhibit homeologous pairing.

The discussion part is somewhat limited, focused on the results of the study, and not proportional to the main manuscript. A connection to similar studies must be attempted. A more detailed comparison of the method with genome-wide association studies (GWAS) for instance must be incorporated.

The incorporation of a flowchart figure illustrating the steps required as a guide for identification and mapping of (a) homozygous and (b) heterozygous mutations could also benefit the manuscript.

Based on the above comments my recommendation is a major revision.

Round 2

Reviewer 2 Report

Dear editor and colleagues,

I have completed my evaluation on the revised version and believe that comments raised were adequately addressed

therefore my recommendation is acceptance of the article (after grammar revision)